# Current Role of Total Laryngectomy in the Era of Organ Preservation

**DOI:** 10.3390/cancers12030584

**Published:** 2020-03-03

**Authors:** Alexandre Bozec, Dorian Culié, Gilles Poissonnet, Olivier Dassonville

**Affiliations:** Institut Universitaire de la Face et du Cou, Centre Antoine Lacassagne, Université Côte d’Azur, 06103 Nice, France; dorian.culie@nice.unicancer.fr (D.C.); gilles.poissonnet@nice.unicancer.fr (G.P.); olivier.dassonville@nice.unicancer.fr (O.D.)

**Keywords:** head and neck cancer, larynx, organ preservation, total laryngectomy

## Abstract

In this article, we aimed to discuss the role of total laryngectomy (TL) in the management of patients with larynx cancer (LC) in the era of organ preservation. Before the 1990s, TL followed by radiotherapy (RT) was the standard treatment for patients with locally advanced LC. Over the last 30 years, various types of larynx preservation (LP) programs associating induction or concurrent chemotherapy (CT) with RT have been developed, with the aim of treating locally advanced LC patients while preserving the larynx and its functions. Overall, more than two-thirds of patients included in a LP program will not require total laryngectomy (TL) and will preserve a functional larynx. However, despite these advances, the larynx is the only tumor site in the upper aero-digestive tract for which prognosis has not improved during recent decades. Indeed, none of these LP protocols have shown any survival advantage compared to primary radical surgery, and it appears that certain LC patients do not benefit from an LP program. This is the case for patients with T4a LC (extra-laryngeal tumor extension through the thyroid cartilage) or with poor pretreatment laryngeal function and for whom primary TL is still the preferred therapeutic option. Moreover, TL is the standard salvage therapy for patients with recurrent tumor after an LP protocol.

## 1. Introduction

Head and neck squamous cell carcinoma (HNSCC, i.e., squamous cell carcinoma arising in the oral cavity, the pharynx, or the larynx) every year accounts for approximately 600,000 new cases worldwide and, in Europe and North America, represents the sixth leading cause of cancer deaths [1]. Most HNSCC, and particularly those occurring in the larynx, representing around 20% of HNSCC, are still related to tobacco consumption [2]. Among larynx cancers (LCs), glottic carcinomas are associated with a lower risk of lymphatic spread and, therefore, a better prognosis than supraglottic carcinomas [1]. Although early dysphonia could, theoretically, allow early diagnosis, LCs continue to be regularly diagnosed at a locally advanced stage and are susceptible, of themselves or due to treatment, to compromised laryngeal functions [3].

Since the 1980s, surgical procedures have drastically evolved from radical and non-conservative to functional and conservative. Several types of partial open laryngectomy have been developed, according to the anatomical subsite and extent of the tumor, thus allowing both appropriate tumor resection and preservation of laryngeal functions (i.e., voice, breathing and swallowing) [4]. More recently, transoral surgery, including transoral laser microsurgery (TLM) and transoral robotic surgery (TORS), has enabled oncologic surgeons to remove early stage LC without opening the upper aerodigestive tract, hence avoiding complications related to conventional open surgery (scar, salivary fistula, infection, etc.) and enabling faster functional recovery, particularly regarding swallowing [5,6]. However, even when the primary tumor is removed by TLM or TORS, neck dissection has to be performed when indicated, and in particular for supraglottic cancer or glottic cancer invading the supraglottic area due to the risk of lymph node involvement [3,6]. At the same time, advances in radiotherapy (RT) techniques and, notably, the development of intensity-modulated radiation therapy (IMRT), have considerably reduced late side-effects of HNSCC RT [7]. Nevertheless, in the specific context of LC patients, there is no study investigating the benefit of IMRT in comparison to conventional RT in terms of laryngeal functions. Some studies have, however, demonstrated a reduction of the average RT doses to the carotid arteries with IMRT in patients with early glottic cancer [8].

Before the 1990s, total laryngectomy (TL), followed by adjuvant RT, was considered to be the gold standard for patients with locally advanced LC. Patients, on the other hand, perceive this procedure as a form of mutilation [9]. However, since TL was first performed, considerable efforts have been made to improve the management of laryngectomized patients by developing voice rehabilitation techniques and prosthetic systems [10]. Since then, the concept of larynx preservation (LP) has gradually emerged, aimed at preserving the larynx without compromising patient survival [11]. Several LP protocols associating RT with induction and/or concomitant chemotherapy (CT) have been developed and tested in randomized prospective clinical trials. Although these complex protocols have led to LP being performed in a significant proportion of cases, none of them have shown any benefit in terms of survival compared to primary radical surgery (i.e., TL) [11,12]. Accordingly, LC is one of the rare human cancers for which survival rates have not increased during the last 30 years [13]. Objective analysis of LP failures and local recurrences in LC patients showed that some patients did not benefit from LP protocols, highlighting the need for careful patient selection. Consequently, there is still an important role for TL in the management of patients with locally advanced LC. 

The aim of this review is to discuss the current role and status of TL in the era of LP. 

## 2. Larynx Preservation Strategies

Most patients with early stage LC (T1–T2 N0 or small N1) can benefit from an LP strategy with either conservative surgery (i.e., partial laryngectomy using the open surgical approach, TLM, or TORS) or RT alone [14]. Locally advanced LC represents a very heterogeneous group of tumors since it includes both T1–T2 tumors with N+ neck and T3–T4 tumors, regardless of the N status. For clarity, we will use the term “locally advanced LC” in this review to refer to advanced T stage (i.e., T3–T4, all N) tumors. Contrary to glottic carcinomas, neck metastases are frequent in T1–2 supraglottic carcinomas [11,12,13,14]. Patients with T1–T2 N+ LC are most often treated by definitive concurrent chemoradiotherapy (CRT), although there is still a role for primary partial laryngectomy with neck dissection followed by adjuvant RT in well-selected cases [11,12,13,14].

LP is a much more challenging strategy in patients with locally advanced (T3–4) LC and has evolved through various types of RT and CT combinations over more than 30 years of clinical trials [11,13]. The negative physical and psychosocial impact of a permanent tracheostomy and the loss of the natural voice are powerful drivers leading patients to favor a treatment that will preserve their larynx [9]. Overall, these LP studies have shown that, in well-selected patients with locally advanced LC and who are candidates for TL, the association of CT and RT can offer LP in a significant proportion of patients without reducing survival rates compared to primary radical surgery [11,13]. However, none of these LP studies showed any survival benefit compared to primary TL and, conversely, there is clearly a subgroup of patients that experience decreased survival and/or poor functional outcomes following an LP therapeutic strategy [13]. This suggests the need for careful patient selection before inclusion in an LP protocol. 

There are two main types of LP strategy (Figure 1). The first, favored in the United States and the UK, is based on RT (66–70 Gy) administered concurrently with three cycles of cisplatin (100 mg/m^2^). The second, favored in most European countries, is based on induction CT followed by RT (± CT or cetuximab) in patients responding to induction CT or by TL and adjuvant RT in non-responders [13]. There is still an ongoing debate worldwide regarding the best LP strategy. The RTOG 91-11 trial, which was updated after a median follow-up of more than 10 years for surviving patients, was specifically designed to assess the contribution of CT added to RT and the optimal timing of CT (induction vs. concurrent) [15,16]. The study enrolled 547 previously untreated patients with locally advanced larynx cancer randomized in 3 treatment groups: CRT, induction CT, and RT alone. The long-term results confirmed those previously published at 5 years showing that the rates of LP were significantly higher for patients in the CRT arm (81.7%). At 10 years, superior outcomes in terms of local control were also shown in favor of concomitant treatment (69.2%, 53.7%, and 50.1%, respectively, for concurrent CT, induction CT, and RT alone). However, no significant difference was found in the composite end point of TL-free survival between the two CT approaches (23.5% and 28.9% for the concurrent and induction CT arms, respectively), which were both superior to RT alone (17.2%). No significant difference was found in the overall survival (OS) among the three treatment arms at 5 and 10 years, although the survival curves did separate after 4.5 years in favor of the induction treatment group (HR, 1.25; *p* = 0.08) compared with the concurrent treatment group. With a long follow-up, the CRT group had the lowest death rate as a result of LC, but a higher rate of non-cancer deaths compared to the induction arm. This outcome could not be explained and, in particular, did not seem to be linked to a higher rate of severe late treatment-related side effects since no difference was found in the cumulative incidence of late toxicities between treatments (grade 3 to 5 range, 31% to 38%), quality of life, or function (swallowing ability, diet, speech, and voice quality), as assessed through the collection of validated questionnaires [15,16]. 

Since the time of the RTOG 91-11 trial, several studies on induction CT in patients with HNSCC have demonstrated improved oncologic outcomes with the addition of docetaxel (T) to the conventional CT regimen cisplatin (P) + 5-fluorouracil (F) [17,18]. One of these studies, conducted by the GORTEC collaborative study group (GORTEC 2000-01 trial), compared the TPF with the PF induction regimen for LP in patients (*n* = 213) with stage III/IV operable larynx/hypopharynx cancer [18]. Responders to three cycles of induction CT (partial response + recovery of larynx mobility) received RT alone and non-responders underwent TL. The initial results presented in 2009 were updated in 2015 after a median follow-up of 8.75 years and showed a significantly higher response rate in the TPF arm compared to the PF arm (80% vs. 59%) [18,19]. The 5- and 10-year LP rates were 74.0% vs. 58.1% and 70.3% vs. 46.5% (*p* = 0.01) in the TPF vs. PF arm, respectively. The 5- and 10-year larynx-dysfunction-free survival rates were 67.2% vs. 46.5% and 63.7% vs. 37.2% (*p* = 0.001), respectively. OS and disease-free survival (DFS) were not statistically improved in the TPF vs. the PF arm. However, statistically fewer grade 3–4 late toxicities of the larynx occurred with the TPF regimen compared with the PF arm (9.3% vs. 17.1%, *p* = 0.04). Finally, this study established the TPF regimen followed by RT as the recommended induction CT-based LP approach [18,19]. 

To further improve LP results using an induction CT strategy, the TREMPLIN randomized phase II study compared cisplatin with cetuximab concurrently associated with RT in responders to three cycles of TPF [20]. Of the 153 enrolled patients, 116 responded to induction CT and were randomly assigned to receive RT + cisplatin (*n* = 60, arm A) or RT + cetuximab (*n* = 56, arm B). In an intent-to-treat analysis, there was no significant difference in LP at 3 months between arms A and B (95% and 93%, respectively), larynx function preservation (87% and 82%, respectively), and OS at 18 months (92% and 89%, respectively). Treatment compliance was higher in arm B, but there were fewer local treatment failures in arm A than in arm B. The authors concluded that there was no evidence that one treatment was superior to the other or could improve the outcome reported in the GORTEC 2000-01 trial with induction CT followed by RT alone [20]. Thus, the protocol that best compares with RT alone after induction CT is still to be determined. 

The two types of LP strategy (i.e., concurrent vs. induction CT) continue to be used for patients with locally advanced LC, with local preferences according to the country and medical centers. The RTOG 91–11 trial did not contain an arm with TPF induction as this trial was initiated before the TPF induction regimen was proved to be superior to PF in the GORTEC 2000-01 trial. There is no published phase III randomized LP study comparing concurrent platinum-based CRT to TPF-based induction CT followed by RT in responders. As a result, there is a need to compare the RTOG concurrent arm with the TPF arm of the GORTEC trial [13]. The ongoing French phase III trial (GORTEC 2014-03-SALTORL, clinicaltrials.gov NCT03340896) is comparing induction TPF followed by RT in responders vs. concurrent cisplatin-based CRT with the composite end point of laryngoesophageal dysfunction-free survival as the primary end-point. Patients with stage T2-3, N0-2 laryngeal, or hypopharyngeal cancer requiring TL are eligible. 

For centers favoring an induction CT-based LP strategy, the criteria chosen to define responders to induction CT represent a critical issue. Although a complete or quasi-complete response was required to pursue conservative treatment in the first LP studies, an objective partial response (response > 50%) was considered to be sufficient when associated with larynx remobilization (restoration of vocal cord mobility) in recent LP clinical trials [11,13,20]. Continuation of non-surgical treatment in poor responders (response < 50%, Figure 2a–c) to induction CT has been associated with poor oncologic outcomes [11,13]. Larynx remobilization is still an important criterion to consider, particularly in patients presenting an objective but incomplete response. The management of patients showing a complete clinical response to induction CT without larynx remobilization is more difficult. Indeed, when the primary tumor has already destroyed the laryngeal nerves and muscles responsible for larynx mobility, complete response to induction CT could be insufficient to restore larynx mobility.

As mentioned previously, not all patients with locally advanced LC will benefit from an LP program. For some patients, and particularly for those with a very infiltrative tumor with extralaryngeal extension through the thyroid cartilage or those with severe laryngeal dysfunction before treatment, primary TL is still the best therapeutic option in terms of oncologic and functional outcomes (Figure 1). 

## 3. Total Laryngectomy as a Primary Therapeutic Option

For patients with extensive LC (gross extralaryngeal extension, large subglottic extension with cricoid cartilage involvement) and/or poor pretreatment laryngeal function (severe impairments of the airway and swallowing, larynx penetrations, and aspirations), better survival rates and QoL are generally achieved with TL than with organ-preservation approaches, and primary radical surgery should be the preferred approach (Figure 1).

### 3.1. T4a Primary Tumors

There are several retrospective studies showing that patients with LC with extralaryngeal extension through the thyroid cartilage (T4a primary tumor, Figure 3) included in an organ-preservation program experienced poor survival and LP rates [13,14,21,22,23]. Chen and Halpern, in a retrospective observational cohort analysis (American National Cancer Database, NCDB) of 7019 patients with stages III and IV LC, showed that the risk of death was similar with CRT and primary TL for stage III tumors but higher with CRT than with primary TL for stage IV tumors [22]. However, interpretation of this study is difficult because advanced overall stage tumors (III and IV) include not only advanced T-stage tumors (T3 and T4) but also early T-stage tumors (T1 and T2) with N+ neck. Grover et al., also using the NCDB, showed with a propensity score analysis that a LP approach for treatment of T4a LC was associated with an increased risk of death (adjusted HR: 1.31; 95% CI: 1.10–1.57, *p* = 0.003) [23]. Gourin et al., in a retrospective single institution chart review of 451 patients with LC, found that for T4 disease, after controlling for operability, primary TL was significantly associated with better OS (55% at 5 years) than either CRT (25%) or RT alone (0%, *p* < 0.001) [24]. Rosenthal et al. reported the experience of the MD Anderson Cancer Center in a retrospective analysis of 221 patients with T4 laryngeal cancer, of whom 73% underwent primary TL and 27% were treated with an LP approach [25]. Although there was no difference in terms of OS, primary TL was associated with a better locoregional control rate and a lower rate of feeding-tube dependency compared with an LP approach [25]. At 5 and 10 years, the rate of patients alive with a functional larynx (no feeding tube, no tracheostomy) was 32% and 13%, respectively [25]. In another retrospective study on 89 patients with T4a laryngeal cancer featuring thyroid cartilage invasion who were treated initially with either TL (*n* = 53) or an LP strategy (*n* = 36), Choi et al. reported that primary surgery was associated with a longer median OS than an LP strategy (87.2 vs. 31.3 months) [21]. Taken together, these results indicate that primary TL should be the preferred therapeutic option for most patients with T4a LC and, particularly, for those exhibiting thyroid cartilage invasion.

Primary TL for patients with T4a LC offers satisfactory oncologic outcomes greatly superior to those reported for patients with T4a hypopharyngeal cancer or those undergoing salvage TL for recurrent LC [26,27,28]. Indeed, Roux et al., in a retrospective monocentric analysis of 63 patients who underwent primary TL, showed a 5-year OS and DSS rates of 56% and 69%, respectively, in the whole cohort, with a significant negative impact of the hypopharyngeal tumor site on DSS (*p* = 0.005; OR, 10.8; 95% CI, 1.9–58.6) [26]. In another retrospective study from the same institution, Milliet et al. showed, on a multivariate analysis, that salvage TL after RT±CT was associated with significantly worse OS and DSS compared with primary TL [28]. 

### 3.2. Irreversible Loss of Laryngeal Functions

Inclusion of a patient in an LP program supposes that the patient will be able to recover laryngeal functions after treatment. However, some patients with very infiltrative primary tumors display irreversible loss of laryngeal function before therapy. There is no absolute criterion to affirm the permanent loss of laryngeal functions. It is therefore a difficult and subjective interpretation that requires great clinical experience and a multidisciplinary discussion. Patients presenting with a very destructive lesion that requires tracheostomy and enteral nutrition due to airway obstruction, larynx penetrations, and aspirations, before starting any treatment, should be considered as poor candidates for LP. These patients will not benefit from an LP program, and primary TL should be recommended even if their primary tumor does not show extra-laryngeal extension [11,13]. In contrast, airway obstruction is not a sufficient criterion to contraindicate LP, in particular when swallowing function is preserved, since certain polypoid tumors can be obstructive without deeply infiltrating and destroying the larynx.

### 3.3. Subglottic Tumor Subsite or Extension

Management of subglottic carcinomas and glottic carcinomas invading the subglottic area remains controversial because these cancers are reputed to be poor candidates for LP. Indeed, due to the proximity of the cricoid cartilage, cartilaginous extension is an early event in subglottic carcinomas, and infraclinic cricoid involvement is common. However, subglottic carcinomas represent less than 5% of LC and there is no published study with a high level of evidence specifically addressing the impact of subglottic involvement on the outcomes of LP approaches. Although T1–2 subglottic carcinomas can be successfully managed by primary RT or CRT depending on N status, the results of LP approaches for T3 lesions are still widely debated [14]. In a retrospective study conducted between 1996 and 2012 on 197 patients with LC invading the subglottic area treated by surgery (62%), RT (18%), or induction CT (20%) as front-line therapy, Levy et al. showed no difference in OS and locoregional control according to front-line treatments [29]. In a multivariate analysis, age > 60 years and positive N stage were the only predictors for OS. Lower locoregional control was observed in T3 patients receiving an LP protocol as compared with those receiving front-line surgery (HR 14.1, 95% CI 2.5–136.7; *p* = 0.02). However, there was no difference in ultimate locoregional control according to the first-line therapy when including T3 patients who underwent salvage TL [29]. Therefore, patients with T3 subglottic carcinoma without evidence of cricoid cartilage extension can be candidates for inclusion in an LP program, whereas patients exhibiting gross cricoid/thyroid cartilage involvement and/or extralaryngeal extension should be referred for primary TL.

### 3.4. Contraindications for a Larynx Preservation Approach

Medical history of neck RT for a previous HNSCC can preclude inclusion in an LP program. Moreover, patients with pretreatment swallowing difficulties and aspirations due to previous HNSCC treatments are not candidates for LP in the event of new primary LC. More debatable is the management of patients with a contraindication for CT due to a poor general health status or advanced age. RT alone is associated with lower oncologic outcomes than CRT in patients with locally advanced LC [15]. Consequently, primary TL should be recommended in patients who accept radical surgery and favor survival. However, RT alone could be offered to fully informed patients for whom LP is the main request. In this latter situation, careful post-RT follow-up is mandatory to avoid delaying salvage TL in the event of treatment failure. In this regard, in a study addressing the impact of medical information delivered to 269 patients faced with a diagnosis of advanced LC amenable to TL or a LP protocol, Laccourreye et al. showed that a total of 28.6% of patients would not consider any trade-off of cure to preserve their larynx and that the median percentage of cure that patients were prepared to lose in order to preserve their larynx was 33% [30]. Almost half the patients wanted to receive additional information before making their decision, with a significant increase among patients with a post-secondary school level of education (*p* = 0.0006) and among those with a family history of cancer (*p* = 0.04). The additional information most frequently requested concerned complications related to the LP protocol (34.1%) and the cure rate (28.6%). After receiving information about the risk of tracheostomy and permanent gastrostomy following the LP protocol, the percentage of subjects who would not consider any trade-off in order to preserve their larynx increased to 31.2% and 56.1%, respectively [30].

## 4. Total Laryngectomy as a Salvage Surgical Procedure

Small recurrences of early stage LC after RT are amenable to partial laryngectomy, either by a transoral (TORS or TLM) or an open surgical approach in well-selected patients [14,31]. Criteria supporting this strategy are preserved preoperative larynx function and, in particular, conserved laryngeal mobility as well as good general health status and pulmonary function. Limited recurrence of early stage laryngeal cancer after primary transoral resection should be preferentially managed by transoral reresection or partial open laryngectomy if a transoral approach is not feasible [13,14]. When salvage conservative surgery is not achievable, RT ± CT could be offered in order to preserve the larynx if laryngeal functions are not compromised by the primary surgical treatment [13,14]. TL should be the preferred salvage therapy when laryngeal functions are already impaired by previous treatment. 

Most patients with recurrence of a locally advanced LC primarily included in an LP program as well as patients with large laryngeal tumor recurrence, whatever the primary tumor stage and treatment, should be managed by salvage TL. Neck dissection must be associated with TL in patients with N+ neck at the time of recurrence. Neck dissection may be avoided in patients with a N0 neck at the time of both the initial treatment and salvage surgery. In this situation, neck dissection could enhance the postoperative complication rate and functional impairment without a demonstrated benefit in terms of survival [13,32]. More controversial is the indication of neck dissection in patients with N+ neck at the time of initial treatment who are N0 (TEP scan evaluation) at the time of laryngeal tumor recurrence. This is a relatively rare situation, and each strategy (neck dissection or not) is associated with several advantages and drawbacks. Neck dissection offers an opportunity to remove possible microscopic residual disease in the lymph nodes but exposes the patient to an increased risk of postoperative complications [13,32,33]. 

Patients with recurrent LC have relatively better survival outcomes than patients with recurrent oral cavity or pharyngeal cancer, in part because the larynx has relatively less robust lymphatic drainage and because the larynx contains natural boundaries that slow local tumor spread [34,35]. Indeed, salvage TL after failure of an LP program in patients with locally advanced LC is associated with 5-year disease-specific survival (DSS) rates generally superior to 50% [34,36]. Li et al., in a retrospective analysis of 100 patients with recurrent LC treated by salvage TL, showed a 5-year DSS rate of 55.2% for patients with a locally advanced tumor on initial tumor staging [36]. In another retrospective study on 244 patients who underwent salvage TL for recurrent LC after RT ± CT, Birkeland et al. reported 5-year OS and DFS rates of 49% and 58%, respectively [37].

## 5. Total Laryngectomy as a Functional Surgical Procedure

A small proportion of patients undergoing partial laryngectomy will not recover laryngeal functions and will experience severe laryngeal aspirations and penetrations [38,39]. Although subclinical larynx aspirations are common, most patients are able to recover oral feeding, and permanent enteral nutrition-dependence rates inferior to 10% are generally reported [39,40]. Indeed, Benito et al., in a retrospective study on 457 patients undergoing supracricoid partial laryngectomy, showed that 41% of patients experienced long-term aspirations, but only 2% of patients required permanent enteral nutrition [39]. Post-operative RT ± CT is a well-known risk factor of long-term swallowing dysfunction after partial laryngectomy [40]. Partial frontolateral laryngectomy with epiglottoplasty (Tucker’s procedure) or supracricoid partial laryngectomy with cricohyoidoepiglottopexy (CHEP) are associated with better swallowing outcomes than supraglottic laryngectomy or supracricoid partial laryngectomy with cricohyoidopexy (CHP) [40]. When permanent enteral nutrition is required despite intensive speech therapy, TL can be offered to patients who primarily want to recover complete oral feeding, after comprehensive information regarding the functional consequences of TL. 

The rate of definitive enteral nutrition in patients included in an LP program is around 5%–10% [13]. In a series of 84 LC patients treated by CRT, Ward et al. showed a rate of severe late dysphagia of 26.5%, with 8 (10%) patients requiring permanent enteral nutrition [41]. Interestingly, five of the eight feeding-tube insertions occurred beyond five years after CRT, thus highlighting the long-term side effects of CRT on larynx functions [41]. In a retrospective chart review of 477 patients undergoing curatively intended treatment for LC, Anschuetz et al. found a 5% rate of feeding-tube dependence at the last follow-up in patients receiving primary CRT [42]. TL can be offered to these patients after adequate patient information in order to restore oral feeding if they accept the loss of laryngeal voice and definitive tracheostoma. Besides permanent enteral nutrition, airway (tracheostomy?) and voice impairments, as well as persistent laryngeal chondroradionecrosis, are also important criteria to consider before discussing TL as a functional procedure [43].

## 6. Functional Results and Quality of Life after Total Laryngectomy

Because of its functional impact and its consequences on patients’ QoL, different types of LP protocols have been used over the past 30 years to reduce the frequency of TL. Indeed, Maddox et al. showed in a retrospective study, conducted in the US, that, between 1997 and 2008, the rate of TL has dropped (−48%) more than the incidence of LC has dropped (−33%), consistent with the trend toward nonsurgical treatment [44]. In most cases, patients undergoing TL recover satisfactory swallowing function [26,28,45]. Long-term feeding-tube-dependence rates between 5% and 10% are generally reported [28,45,46]. Better swallowing outcomes are achieved for larynx than for hypopharynx cancer patients [26,29]. However, some laryngeal tumors with posterior or posterolateral extension to the retrocricoid region or the piriform sinus can require pharyngeal resection similar to that of hypopharynx tumors. In a study on 63 patients undergoing primary TL for T4 laryngeal/hypopharyngeal cancer, Roux et al. showed that a normal/subnormal swallowing (normal consistency oral diet) function was recovered 6 months after surgery in 87% and 67% of patients with laryngeal and hypopharyngeal cancer, respectively [26]. When TL includes large pharyngeal resection (circular or near-circular defect), the loss of pharyngeal motor and sensitive functions and the frequent development of pharyngo-esophageal stricture, particularly after adjuvant RT, explain the difficulties in recovering good swallowing function [26,47]. 

With the development of the tracheoesophageal prosthesis (TEP), most laryngectomized patients are able to recover a functional voice [10,45]. Success rates for voice rehabilitation with TEP approximating 85% have been reported by several authors and are superior to those reported with esophageal speech [10,45]. Whatever the speech rehabilitation method used, pharyngo-esophageal stricture, chronic bronchitis/poor pulmonary function, high comorbidity level neurological/psychological impairments, and lack of motivation have been associated with a higher risk of voice rehabilitation failure [10]. Besides their high success rates, TEP can claim numerous advantages, including quasi-immediate voice production and production of sustained more fluent quality of speech than with esophageal speech [48]. However, TEP has a mean lifetime of around 3 to 4 months with high lifetime variability, requires regular replacement, and is associated with frequent minor complications, which, although generally easily managed, can be constraining for the patients [10,49,50]. Severe complications are rare but can be particularly difficult to manage [10,51]. This is the case for major enlargements of the tracheo-esophageal fistulae that occur in less than 5% of patients but can require highly complex surgical reconstruction [52,53]. However, with an experienced multidisciplinary team, the advantages of using TEP far outweigh their drawbacks [50,54]. Currently, there is no consensus that exists with regard to the timing of performing TEP, and the decision to perform a primary (at time of TL) or secondary TEP has mostly been based on physician preference [10,55]. In a recent systematic review, Luu et al. showed no significant differences between primary and secondary TEP in terms of complications or voice outcomes [55]. Due to an easier technical procedure and faster voice rehabilitation, we believe that primary TEP should be favored for most patients except those in poor general conditions or those undergoing extensive salvage TL (TL with circular or quasi-circular pharyngectomy or TL with large tracheal resection) [10]. Indeed, peristomal and fistula wall necrosis occurred more frequently when primary TEP was performed in previously irradiated patients undergoing this type of TL [10]. 

Several studies have investigated QoL of laryngectomized patients. Surprisingly, higher-than-expected QoL scores have been regularly reported, suggesting that, overall, long-term QoL seems satisfactory for most patients [50,56,57,58]. Compared with patients undergoing the LP protocol, laryngectomized patients do not report worse global QoL scores. However, significant differences have been reported in some specific QoL domains [58,59]. Sanabria et al., for instance, showed no statistically or clinically significant differences in the global QoL score between TL and LP protocols [58]. The most important domains affecting QoL for both groups were speech and activity. In the TL group, the next most important domains were appearance, taste, pain, and recreation, whereas, in the LP group, they were saliva, recreation, mood, and swallowing [58]. In an interesting study assessing voice and swallowing outcomes of an LP trial, Fung et al. showed that mean voice-related QoL scores of the patients were lower compared with normative data (*p* < 0.05) but higher for LP patients compared with patients who underwent salvage TL (*p* < 0.05) [60]. The understandability of speech was lower in laryngectomized patients (*p* = 0.001), whereas eating in public and normalcy of diet were similar between the two groups [60]. In a study evaluating voice-related QoL after LC treatment, Oridate et al. found that laryngectomized patients reported lower scores than those included in an LP program (mean score: 68.4 vs. 85.5) [61]. Consistently, they also showed higher Voice Handicap Index (VHI)-10 scores (reflecting higher voice problems) in patients who underwent TL than in those included in an LP protocol (mean score: 11.26 vs. 5.43) [61]. In another comparative study between TL and LP protocols, Hanna et al. found no statistically significant differences in the overall QoL score between the two groups [59]. Laryngectomized patients reported significantly greater difficulties with sensory disturbances (smell and taste), use of painkillers, and coughing. On the other hand, patients included in an LP protocol reported significantly greater problems with dry mouth. Interestingly, in this study, there was no statistically significant difference regarding speech problems between laryngectomized patients (mean score: 29.7 ± 23.6) and patients included in an LP program (mean score: 22.2 ± 26.6) [59]. Despite these results showing acceptable QoL scores, it must be kept in mind that TL is still considered to be mutilating for patients experiencing a major impact in their daily life due to the physical and functional sequelae [9]. Psychosocial consequences are also important and impact both patients and their next of kin [62,63]. A functional rehabilitation protocol and an optimal therapeutic education program need to be integrated into the multidisciplinary management to further improve the QoL of laryngectomized patients [62,63]. Among the promising functional rehabilitation methods, silent speech interfaces (SSIs), which represent a novel technological paradigm, have the potential to provide an alternative way to assist laryngectomized patients to produce speech with natural-sounding voices from the movements of their articulators, such as the tongue and lips [64]. SSIs typically include an articulatory movement recorder, a silent speech recognizer, and a text-to-speech synthesizer. This technology could offer the opportunity to significantly improve the voice-related QoL of patients undergoing TL [64].

## 7. Conclusions

Over the last 30 years, the progress made in LP protocols associating induction or concurrent CT with RT has enabled more than two-thirds of patients with locally advanced LC to preserve their larynx. However, none of these LP protocols have shown any survival advantage compared to radical surgery. Consequently, there is still an important role for TL in patients with locally advanced LC. TL is recommended as the primary treatment modality for patients with T4a LC. Primary TL should also be considered for T3 LC patients with severe and irreversible impairment of laryngeal function, with large subglottic extension invading the cricoid cartilage, or with contraindication to an optimal LP protocol (previous RT to the neck, contraindication to CT, advanced age, poor general condition). Moreover, salvage TL is the standard treatment for recurrent LC after the failure of organ preservation treatment. Multidisciplinary management of laryngectomized patients integrating intensive functional rehabilitation and a personalized therapeutic education program needs to be established to improve QoL and psychosocial well-being. 

## Figures and Tables

**Figure 1 cancers-12-00584-f001:**
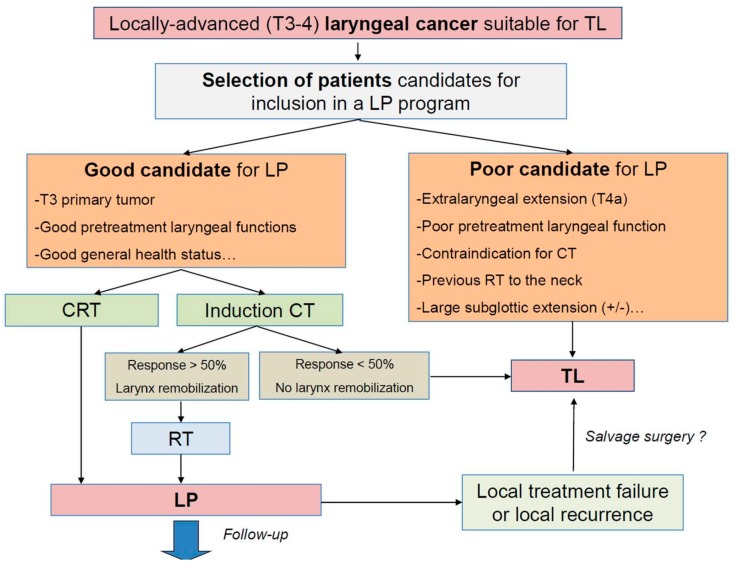
Total laryngectomy and larynx preservation strategies in patients with locally advanced laryngeal (T3–4) cancer. TL: total laryngectomy; LP: larynx preservation; RT: radiotherapy; CT: chemotherapy; CRT: concurrent chemoradiotherapy.

**Figure 2 cancers-12-00584-f002:**
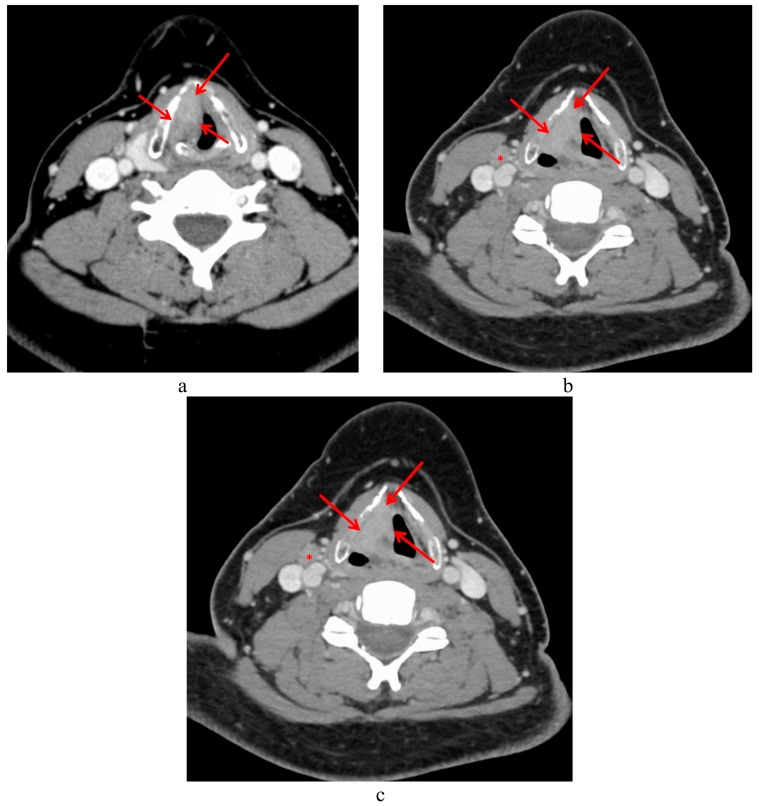
CT scan of a 36-year-old patient with T3 laryngeal cancer (red arrows) involving the right glottic and supraglottic levels with laryngeal immobility, and a left metastatic lymph node in level III (red asterix). Indication of inclusion in a larynx preservation program. Pretreatment CT scan (**a**,**b**) and CT-scan after three cycles of induction chemotherapy showing a poor response to therapy (**c**). Indication of total laryngectomy for induction chemotherapy failure.

**Figure 3 cancers-12-00584-f003:**
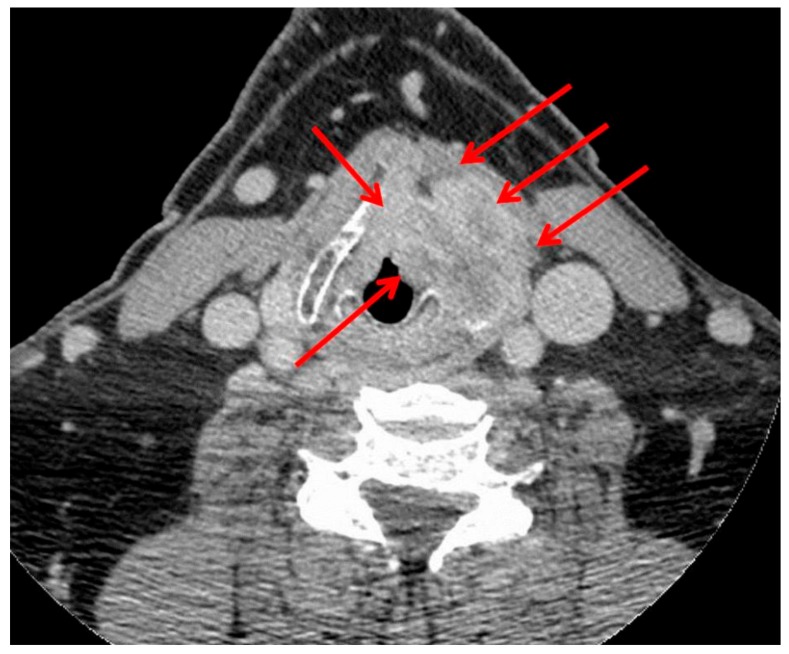
CT scan of a 72-year-old patient with T4a laryngeal cancer (red arrows) showing a large extralaryngeal extension through the left side of the thyroid cartilage. Indication of primary total laryngectomy.

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
