# Peer review of "Current Role of Total Laryngectomy in the Era of Organ Preservation"

_cancers, 2020, doi:10.3390/cancers12030584_

Round 1

Reviewer 1 Report

The manuscript discussed the role of total laryngectomy (TL) in the management of patients with laryngeal cancer (LC) in the era of organ preservation, which is interesting. The discussion focused on the larynx preservation (LP) program in the past decades.  No significant difference was found in terms of survival advantage over the two groups (TL and LP). On the other hand, the quality of life (e.g., swallowing and speech) of laryngectomized patients have shown higher score that have been regularly reported.  Compared with patients undergoing the LP protocol, laryngectomized patients do not report worse global QoL scores. Thus, TL is still recommended as the primary treatment modality for patients with T4a or T3 LC.  Overall, this manuscript is well written and informational.  A few minor to moderate improvement can be done before published.

First, more detailed discussion about the QoL (particular speech) between patients with TL or LP could be added in Section 6.  Since the major conclusion is that there is no significant difference in terms of survival rate between them.  The QoL of with or without participating LP became more interesting and critical in helping the patients’ decision making if they would like to preserve the larynx or not.  The current manuscript discussed speech/voice, which is good.  A more detailed comparison on the speech/voice quality (including both advantages and complications) of these two groups of patients that may also affect their social life may be even better. The authors also mentioned the concerns secondary to speech were different between the two groups. For example, TL group concerns more on appearance, taste, pain, etc., while the LP groups concerns more on saliva, recreations, mood, and swallowing.  More details (e.g., one to one comparison between the two groups) would be helpful.

It may be good to mention an emerging technology called silent speech interface (SSI) is under development, which could be also in section 6.  The idea of SSI is to output real-time synthesized, high intelligible speech based on the oral movements (e.g., tongue and lip movements), which holds potential to assist the oral communication of the TL group.  SSI may be helpful for LP group as well. Recent break-through has been made based on the data collected from healthy speakers using permanent magnetic articulography (PMA), electromagnetic articulograph (EMA), ultrasound, sEMG for oral movement tracking.  Once successful, SSI will significantly improve the quality of  life of these patients by improving their speech.

Denby, T. Schultz, K. Honda, T. Hueber, J. M. Gilbert, J. S. Brumberg, "Silent speech interfaces", Speech Communication, vol. 52, pp. 270-287, 2010.

Kim, M., Cao, B., Mau, T., & Wang, J. (2017). Speaker-independent silent speech recognition from flesh point articulatory movements using an LSTM neural network, IEEE/ACM Transactions on Audio, Speech, and Language Processing, 25(12): 2323-2336.

S. Meltzner, J. T. Heaton, Y. Deng, G. De Luca, S. H. Roy and J. C. Kline, "Silent Speech Recognition as an Alternative Communication Device for Persons With Laryngectomy," in IEEE/ACM Transactions on Audio, Speech, and Language Processing, vol. 25, no. 12, pp. 2386-2398, Dec. 2017.

Cao, B., Sebkhi, N., Mau, T., Inan, O. T., & Wang, J. (2019). Permanent magnetic articulograph (PMA) vs electromagnetic articulograph (EMA) in articulation-to-speech synthesis for silent speech interface, Workshop on Speech and Language Processing for Assistive Technologies, pp. 17-23.

Gosztolya, Á. Pintér, L. Tóth, T. Grósz, A. Markó and T. G. Csapó, "Autoencoder-Based Articulatory-to-Acoustic Mapping for Ultrasound Silent Speech Interfaces," International Joint Conference on Neural Networks (IJCNN), 2019, pp. 1-8.

The second is about writing, which is minor.

For Abstract, personally I suggest to move the last sentence to the very beginning. 

Figure 1, the “LT” in the middle-right box should be “TL”.

An indicator (e.g., an arrow or circle) can be added to show where exactly to look at in Figures 2 and 3. Readers who are not familiar with the CT images may have difficulty what to look at and how to interpret them.

Figure 2’s caption, more detailed explanation will be helpful, particularly what’s in (a), (b), and (c), respectively.

Last,  some important and relevant references are missing and they can be added.

Maddox PT, Davies L., Trends in total laryngectomy in the era of organ preservation: a population-based study, Otolaryngol Head Neck Surg. 2012 Jul;147(1):85-90.

Raol N, Hutcheson KA, Lewin JS, Kupferman ME. (2013). Surgical Management of the Non-functional Larynx after Organ Preservation Therapy. J Otol Rhinol 2:1.

Author Response

Dear Editor,

Dear Reviewers,

We are grateful to the Editorial Board and to the experts for considering our submitted manuscript entitled “Current role of total laryngectomy in the era of organ preservation”. We have taken into account all remarks and criticisms formulated by the reviewers and we thank the Editorial Board and the experts for their constructive comments. The changes are clearly highlighted in the text (in red) and our answers are as follows:

Reviewer 1:

  1. “First, more detailed discussion about the QoL (particular speech) between patients with TL or LP could be added in Section 6… More details (e.g., one to one comparison between the two groups) would be helpful.”

- We have provided more details regarding QoL and voice outcomes between patients with TL or LP (see section 6). Two new references focusing on this topic have been added and discussed.

  1. “It may be good to mention an emerging technology called silent speech interface (SSI)… Once successful, SSI will significantly improve the quality of life of these patients by improving their speech.”

-We have mentioned this promising technology at the end of the Discussion section and added a new reference on this topic ([64]). 

  1. For Abstract, personally I suggest to move the last sentence to the very beginning.

-This has been done.

  1. Figure 1, the “LT” in the middle-right box should be “TL”.

-This has been done.

  1. An indicator (e.g., an arrow or circle) can be added to show where exactly to look at in Figures 2 and 3. Readers who are not familiar with the CT images may have difficulty what to look at and how to interpret them.

Figure 2’s caption, more detailed explanation will be helpful, particularly what’s in (a), (b), and (c), respectively.

-This has been done.

  1. Last, some important and relevant references are missing and they can be added.

-These 2 references ([43] and [44]) have been added.

Reviewer 2 Report

This is a useful practical summary of management of advanced laryngeal cancer , discussing laryngeal preservation v laryngectomy in terms of oncological and functional outcomes.

Some minor comments:

Intro

Bit confusing re. larynx cancer  in 5th line – comparing prognosis of Larynx cancer in one line, and in the next contrasting with supraglottic cancer.  I think this is a comparision of glottic v supraglottic but should be clearer.

2nd paragraph – discusses transoral surgery – comments relate mainly to glottic cancer.  Since this review of is larynx cancer not just glottic it needs to be clear that supraglottic cancers would need treatment of neck due to nodal risk.

Re. IMRT reducing late effects of RT – is there any evidence of this for larynx cancer?  If so, useful to mention. If not, useful to say as it is not clear that IMRT has really reduced late toxicity of laryngeal treatment.

Larngeal preservation section

Need to define/discuss what is meant ‘locally advanced LC’ – eg. of course can be T3/4 but also can be early T stage with N+ neck.  Ie. The locally advanced group is very heterogenous

Figure1 : does not include whole range of ‘locally advanced LC’ – only T3/4. What about T1/2N+. This would be rare for glottic but much more common for supraglottic cancers. Again the review is about larynx cancer not just glottic.

Also in Figure 1 the box CT+RT is confusing  - I think it refers to concurrent – better written as ‘CRT’ or ‘Conc CRT’..

Is it possible to say when the GORTEC 2014-03-SALTORL trial is due to close/report?

Please try to define ‘remobilisation’ – is this referring to cord mobility?

Section 3:

1st line – please try to explain what is meant by ‘extensive’ LC and by ‘poor pretreatment laryngeal function’  more clearly.

Re. ref 21 – what are the stage IV cancers – are they advanced T stage or do they include early T stage with advanced nodal disease ? eg with supraglottic cancers.

Section 3.2 – do the authors have any insights into what degree of functional loss is likely to be irreversible? Eg. if needing tube feeding? What about silent aspiration? Poor voice quality?

What about patients presenting with airway problems and end up with debulking – are they likely to have irreversible loss of function? 

The authors views on these different functional aspects and their potential for recovery or otherwise with LP would be useful.

Section 3.3 is useful discussing subglottic disease.  It seems odd there is not a corresponding section for supraglottic primaries or supraglottic extension.

Section 6: It would be useful to consider the merits of primary or secondary valve puncture. Ie. Creating valve at time of TL or performing a secondary puncture after adjuvant radiotherapy.

Author Response

Dear Editor,

Dear Reviewers,

We are grateful to the Editorial Board and to the experts for considering our submitted manuscript entitled “Current role of total laryngectomy in the era of organ preservation”. We have taken into account all remarks and criticisms formulated by the reviewers and we thank the Editorial Board and the experts for their constructive comments. The changes are clearly highlighted in the text (in red) and our answers are as follows:

Reviewer 2:

  1. “Intro: Bit confusing re. larynx cancer in 5th line – comparing prognosis of Larynx cancer in one line, and in the next contrasting with supraglottic cancer.  I think this is a comparison of glottic v supraglottic but should be clearer.”

-We have clarified this paragraph (see Introduction section)

  1. “2nd paragraph: discusses transoral surgery – comments relate mainly to glottic cancer. Since this review of is larynx cancer not just glottic it needs to be clear that supraglottic cancers would need treatment of neck due to nodal risk.”

-We have clarified this paragraph (see Introduction section) and stipulated that supraglottic cancers need treatment of the neck

  1. “IMRT reducing late effects of RT – is there any evidence of this for larynx cancer? If so, useful to mention. If not, useful to say as it is not clear that IMRT has really reduced late toxicity of laryngeal treatment.”

-We agree that is not clear that IMRT has really reduced late toxicity of RT in terms of laryngeal functions. However, a reduction of the average RT doses to the carotid arteries is well demonstrated. This has been clarified in the Introduction section.

  1. “Need to define/discuss what is meant ‘locally advanced LC’ – eg. of course can be T3/4 but also can be early T stage with N+ neck. The locally advanced group is very heterogenous”.

We agree that the term “locally advanced LC” is confusing and we have discussed this issue in section 2.

  1. “Figure1: does not include whole range of ‘locally advanced LC’ – only T3/4. What about T1/2N+. This would be rare for glottic but much more common for supraglottic cancers. Again the review is about larynx cancer not just glottic. Also in Figure 1 the box CT+RT is confusing - I think it refers to concurrent – better written as ‘CRT’ or ‘Conc CRT’”.

-We have briefly discussed the management of patients with T1-2 N+ LC in the “larynx preservation strategies” section. We have modified the legend and title of Figure 1 to clarify that this figure illustrates the management of patients with T3-4 LC suitable for TL. We have changed the term “CT+RT” which is confusing for CRT in Figure 1 and in the text.

  1. “Is it possible to say when the GORTEC 2014-03-SALTORL trial is due to close/report?”

-This study is still recruiting. The inclusion period was recently extended to reach the number of patients to be recruited. I am not able to provide more information about the end of this study in this review.

  1. “Please try to define ‘remobilization’ – is this referring to cord mobility?”

-We have defined “larynx remobilization” that refers to restoration of vocal cord mobility.

  1. “1st line – please try to explain what is meant by ‘extensive’ LC and by ‘poor pretreatment laryngeal function’ more clearly”

-We have explained more clearly what is meant by extensive LC and poor pretreatment laryngeal function in section 3.

  1. “Re. ref 21 – what are the stage IV cancers – are they advanced T stage or do they include early T stage with advanced nodal disease ? eg with supraglottic cancers”.

-We agree that this study is difficult to interpret because the impact of overall stage (rather than T-stage) on OS was considered. We have discussed this issue in section 3.1.

  1. “Section 3.2 – do the authors have any insights into what degree of functional loss is likely to be irreversible? Eg. if needing tube feeding? What about silent aspiration? Poor voice quality?”What about patients presenting with airway problems and end up with debulking – are they likely to have irreversible loss of function? The authors views on these different functional aspects and their potential for recovery or otherwise with LP would be useful.

-There is no absolute criterion to affirm the permanent loss of laryngeal functions. It is therefore a difficult and subjective interpretation that requires a great clinical experience and a multidisciplinary discussion. This issue has been discussed in the section 3.2.

  1. “Section 3.3 is useful discussing subglottic disease. It seems odd there is not a corresponding section for supraglottic primaries or supraglottic extension.”

-We believe that subglottic LC pose particular problems because the rate of cricoid cartilage invasion is high and is associated with a high risk of LP failure even in the absence of extra-laryngeal extension.

  1. “Section 6: It would be useful to consider the merits of primary or secondary valve puncture. Ie. Creating valve at time of TL or performing a secondary puncture after adjuvant radiotherapy”.

-We have discussed this issue in section 6.